# Local Wisdom of West Timorese Farmers in Land Management



Yohanis Ngongo [1,*], Tony Basuki [1], Bernard deRosari [1], Evert Y. Hosang [1], Jacob Nulik [1], Helena daSilva [1], Debora Kana Hau [1], Alfonso Sitorus [1], Noldy R. E. Kotta [1], Gerson N. Njurumana [2], Eko Pujiono [2], Lily Ishaq [3], Agnes V. Simamora [3] and Yosep Seran Mau [3]

[1] East Nusa Tenggara Assessment Institute of Agricultural Technology (ENT AIAT), Kupang 85362, East Nusa Tenggara, Indonesia; tony.basuki84@gmail.com (T.B.); benderosari@yahoo.com (B.d.); yulianeshosang@yahoo.co.id (E.Y.H.); jacob_nulik@yahoo.com (J.N.); helena_dasilva73@yahoo.com (H.d.); debora_nulik@yahoo.com (D.K.H.); sitorusalfonso@gmail.com (A.S.); noldy_kotta@yahoo.com (N.R.E.K.)

[2] Research Center for Ecology and Etnobiology, National Research and Innovation Agency (NRIA), Jl. Raya Jakarta-Bogor Km. 46, Cibinong 16911, West Java, Indonesia; gers001@brin.go.id (G.N.N.); eko.pujiono@brin.go.id (E.P.)

[3] Agriculture Faculty, University of Nusa Cendana, Kupang 85001, East Nusa Tenggara, Indonesia; i-ishaq@staf.undana.ac.id (L.I.); asimamora@staf.undana.ac.id (A.V.S.); yosepmau@staf.undana.ac.id (Y.S.M.)

[*] Correspondence: yohanisngongo@gmail.com; Tel.: +62-813-5329-3979

**Abstract:** This paper's working hypothesis is that the indigenous farming practices of Timorese farmers are those most suitable and adaptable with regard to these farmers' circumstances. Intensive farming and the acceleration of land conversion in Java lead to a reduction in favorable cropland and the degradation of soil biology. To meet the demand for food production, unfavorable areas outside Java, including marginal semi-arid areas on Timor Island, East Nusa Tenggara province, have become an important option. Unfortunately, the national crop production policy has paid less attention to the specific biophysical characteristics of the region and how local people have adapted to the diverse marginal environment. We review the literature in the areas of soil nutrition retention and soil biology, vegetation/crop diversity, and farming practices/management, including local wisdom on soil management. This paper highlights that the values of the chemical parameters of the soils in question are varied, but generally range from low to high. The existence of beneficial micro-organisms is important both for improving soil fertility and due to their association with local vegetation/crops. Traditional farming practices, such as the local agroforestry of Mamar, have effectively preserved the existence of micro-organisms that promote conservation practices, crop/vegetation diversity, and sustainable agriculture. We recommend that the expansion of croplands and crop production into marginal semi-arid areas needs to be considered and adapted while taking into consideration sustainability and environmentally sound traditional practices.

**Keywords:** farming system; natural vegetation; local wisdom; marginal areas; soil biology; Timor Island

## 1. Introduction

Increasing food crop production is still a major challenge for agricultural development in Indonesia. The production of food crops, particularly rice as a main staple, mainly relies on the unfavorable cropland in Java or in the western parts of Indonesia in general. Moreover, rice production policy is biased towards subsidies for agro-chemical inputs, particularly fertilizers [1]. Meanwhile, rice productivity in favorable areas in Java has been reported to be stagnant [2], and the current national rice production is no longer keeping pace with the increasing human population [3].

Unfavorable cropland areas outside Java, including the dominant semi-arid area of East Nusa Tenggara province, have become important options. Nevertheless, food crop production, particularly rice in ENT, remains unchanged. The ENT province still relies on inter-island rice imports of around 100,000 tons per year to meet domestic demand [4].

This figure indicates that implementing the principles of the national food program's bias towards agro-chemical inputs in semi-arid areas has little impact on food crop production. Indeed, agricultural programs in general have negative effects on the environment, including harming biodiversity [5] and soil biology [6,7].

The characteristics of the mosaic environment, including the diversity of vegetation/plants and local micro-organisms, are manifestations of local soil and climatic conditions [8]. For example, in the case of the *Mamar* Agroforestry agro-ecosystem environment, which was studied at 16 different sites, the analysis results indicate that this agro-ecosystem is an important habitat for 112 plant species [9]. There are even several local genetic resources that have been identified and registered as local varieties at the Center for Plant Varieties Protection and Licensing, Ministry of Agriculture, including corn, beans, fruits, spices, and animal feed [10].

Facing the marginal land conditions of West Timor, adaptation efforts by farmers related to food production and planting activities have led to various local knowledge and practices, including the existence of agricultural commodities that are derived from natural selection, which have been ongoing for quite some time. Other local wisdom, which relates to improving soil fertility or maintaining soil health, includes mixed planting with legumes as feed sources, such as planting Lamtoro Taramba (*Leucaena leucocephala* (Lam.) de Wit) or butterfly pea *(Clitoria ternatea)* [11], together with various nuts in one hole or so-called "Sen" [12]. Other examples include using the residues of plants such as bamboo [13], as well as the use of livestock manure from cattle barns or taken from pastures. All of these practices do not introduce external inputs.

Despite the positive aspects of employing local wisdom in farming practices, there are also farming practices that are considered counter-productive and threaten the survival of local micro-organisms and soil decomposers that benefit soil fertility. Examples of these counter-productive practices include the slash-and-burn clearing of land that is carried out before planting. Although the impact of fire on micro-organisms depends on temperature, duration, and soil moisture [14,15], the varying response of micro-organisms affects the soil recovery process [14,16,17]. The short-term impact of fire (measured a day after the fire) was found to include a decline in microbial C biomass, enzyme activities, soil respiration, and microbial activities [17,18]. However, two weeks after the fire, the levels of bacteria, soil respiration, and fungi biomass increased.

Efforts to improve soil fertility and health by combining the potential of local micro-organisms and existing local practices will trigger an increase in agricultural production by farmers. It is strongly suspected that this combination strategy will be easily adopted by farmers because it is in the familiar context of low external inputs. This article is the result of a review of various research that has been carried out in West Timor, as well as other sources of information that have relevance to the land conditions and farming systems in West Timor. The emphasis in this article is a description of soil fertility and health, the diversity of vegetation/plants, the role of farmers, and local practices that allow for adaptation to the challenges of the farming environment. The results of this review describe the role of farmers in adapting to the marginal soil environment; their approaches are expressed in the form of local wisdom that is different from that of other places.

## 2. Literature Review

### 2.1. Soil Nutrient Retention and Its Availability

Climatologically, the western part of Timor Island, East Nusa Tenggara (ENT) province, Indonesia, is classified as a semi-arid area, with a limited amount of annual rainfall; the precipitation is less than 1500 mm/year, and there are only three to four wet months per year, namely, December to March/April [19]. This differs from the dominant wet tropical climate of the western part of Indonesia. Another feature is that most of the soils are young; they are characterized by a thin solum depth of <20 cm and a sloping land surface that is due to the topography, which is dominated by hills and mountains [20]. Topographic conditions show a positive correlation with soil moisture [21]. In terms of the

soil fertility of West Timor, the soil reaction (pH) is alkaline, but there are still problems with the availability of nitrogen and phosphate and the low C-organic content of the soil. This condition is a challenge for agricultural production faced by local farmers and the government in the expansion of agricultural development programs. The consequence of this challenging situation is that without being supported by sound knowledge and cultivation technology, agricultural productivity is low, and there is at times no harvest.

The altitude of West Timor is dominated by the lowland area of below 700 m above sea level (m asl), which covers 86% of the highland area above 700 m asl [19] (Figure 1). In terms of land development, West Timor is dominated by young soils, such as Entisols and Inceptisols, as well as more developed soils, such as Vertisols and Alfisols. These soils develop both from calcareous sediments and from deposit parent materials [20]. In the perspective of soil nutrient retention and nutrient availability, the values of soil chemical parameters are varied, but generally lie in the low to high category.

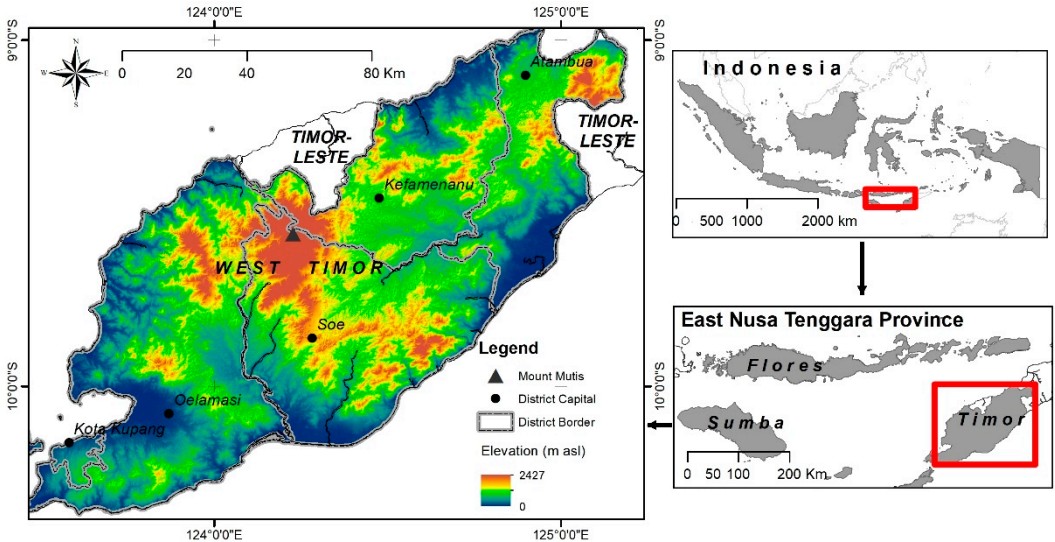

**Figure 1.** West Timor, based on elevation ranging from lowland to highland.

Regarding the presence of C-organic, it can be described that it almost varies between locations. In [22], the soils overgrown with Sandalwood (*Santalum album* L.) in the North Central Timor district contained 1.18% (low) and 4.39% (high) organic C-organic. Meanwhile, as reported by [23], from each type of Vertisol and Alfisol soil in the Kupang district, the C-organic values were 1.26 and 1.05% (low), respectively, including rice land in the Malaka district [24]. Similar conditions were also found in other locations in Kupang [25] in the medium category (2.85%) [26].

In terms of Cation Exchange Capacity (CEC), it is generally classified as medium to high, ranging from 19.84 to 31.67 cmol(+) kg$^{-1}$ [25,27]. The existence of a good CEC value is closely related to the area that is dominated by clay type 2:1 montmorillonite [28]. The characteristics of nutrient retention, such as acidity (pH H$_2$O), C-organic, and CEC, mentioned above, are not much different from the results observed at 16 locations in the agroforestry environment in West Timor, as presented in Table 1. Table 1 shows that the value of Base Saturation (BS) is high (100%). Thus, from the perspective of soil fertility, the nutrient retention aspect is considered quite good.

**Table 1.** The range of minimum, maximum and average values of soil chemical parameters, at 16 observation sites in West Timor.

| Soil Chemical Parameters | Minimum | Maximum | Average | Category * |
|---|---|---|---|---|
| pH (H$_2$O) | 7.10 | 8.20 | 7.83 | Slightly alkaline |
| pH (KCl) | 6.10 | 7.90 | 7.11 | Neutral |
| C (%) | 1.53 | 8.79 | 3.07 | High |
| N (%) | 0.13 | 0.47 | 0.23 | Medium |
| P-potential (P$_2$O$_5$ HCl 25%) (mg/100 g) | 11.00 | 155.00 | 76.44 | Very high |
| K potential (K$_2$O HCl 25%) (mg/100 g) | 25.00 | 316.00 | 105.75 | Very high |
| P-available (P$_2$O$_5$ Olsen) (ppm) | 8.00 | 45.00 | 17.19 | High |
| Exchangeable Ca (cmol(+)/kg) | 27.14 | 52.00 | 41.35 | Very high |
| Exchangeable Mg (cmol(+)/kg) | 0.39 | 6.24 | 2.20 | High |
| Exchangeable K (cmol(+)/kg) | 0.11 | 1.97 | 0.65 | High |
| Exchangeable Na (cmol(+)/kg) | 0.10 | 0.31 | 0.24 | Low |
| Cation Exchange Capacity (cmol(+)/kg) | 11.26 | 44.76 | 24.51 | High |
| Base Saturation (%) | 100.00 | 100.00 | 100.00 | Very high |

Note: * Categories were classified based on [29]: pH H$_2$O: slightly alkaline (7.6–8.5); pH K$_2$O: neutral (6.6–7.5); C: high (3–5%); N: medium (0.21–0.50%); P-potential (P$_2$O$_5$ HCl 25%): very high (>60 mg/100 g); K potential (K$_2$O HCl 25%): very high (>60 mg/100 g); P-available (P$_2$O$_5$ Olsen): high (16–20 ppm); exchangeable Ca: very high (>20 cmol(+)/kg); exchangeable Mg: high (2.1–8.0 cmol(+)/kg); exchangeable K: high (0.6–1.0 cmol(+)/kg); exchangeable Na: low (0.1–0.3 cmol(+)/kg); CEC: high (25–40 cmol(+)/kg); base saturation: very high (>80%). Source: [30].

Regarding the nitrogen status in Timor, some places are in the range of 0.26–0.36% (medium category) [26,27]. Furthermore, the potential P (Phosphate) and K (Potassium) statuses are in the very high category, while the available P is classified as low, as presented in Table 1. Meanwhile, at other locations, the available P status varied from very low to very high [22,26,27,31]. Thus, in the context of nutrient availability, soils in West Timor are quite varied in their availability, ranging from very low to moderate. Therefore, the handling of nutrient availability warrants further attention.

Efforts to increase the availability of P can be made, among others, through the inoculation of phosphate solubilizing microorganisms (PSMs). Some microorganisms that can be utilized to increase P in semi-arid environments, among others, are *Pantoeaa glomerans* [32], *Rhizobium* sp. [33], *Bacillus subtilis* [34], *Penicillium* spp., *Aspergillus* spp., and *Penicillium* spp. [35]. These microorganisms secrete organic acids and make it possible for P to dissolve crops [36]. The application of organic matters and the inoculation of phosphate dissolved microorganisms will boost the inoculation effectivity of available P [37], which is in line with [38], showing that soil organic matter has a positive correlation with P availability and the amount of dissolved P microorganism, and it has a negative correlation with P total.

### 2.2. Soil Biology in West Timor

Soil biology is one of the major soil properties that are often considered an indicator of soil health. Two important indicators commonly used to assess soil biology properties include soil organic carbon and soil microorganisms. Soil organic carbon (SOC) plays an important role in maintaining physical, chemical, and biological properties in the soil, and therefore, the SOC has been recognized as the single most significant soil health indicator [39–41]. Generally, soil organic carbon in West Timor ranges from low to moderate, but mostly, the SOC is low. The low content of SOC in West Timor is very likely related to the low rainfall in the region, which limits plant growth and the deposit of plant residues in the soil, and the hot weather, which favors the decomposition process of organic matter.

Beneficial soil microorganisms, in particular, are involved in many crucial processes in the soil and in the natural fertility of soil [42,43]. For example, a non-functional group of soil microorganisms, often called the decomposer, plays an important role in the mineralization process of organic matter, leading to the release of nutrients in the soil ready for plant uptake, and a functional group of soil microorganisms, which specialize in a certain function in the soil or for plant growth. Examples of these are free living and symbiotic N$_2$-fixing bacteria, mycorrhizas, plant-growth-promoting rhizobacteria (PGPR), and *Trichoderma*.

These beneficial soil microorganisms are important in supporting plant growth in West Timor where the soil is commonly less fertile, and where water becomes a limiting factor for plant growth.

Few studies have been undertaken on the potential of beneficial soil microorganisms in West Timor. The genetic biodiversity of leguminous plants in West Timor, such as wild legumes or legumes, cultivated around the farm land for food or for forage, has been explored [44]. Regarding the low fertility of the soil in West Timor, the occurrence of these leguminous plants is very important due to their role in contributing nitrogen (N) to the soil through the activity of Rhizobium in the roots. The occurrence of ingenious beneficial microorganisms involved in N fixation has also been reported [45].

The presence of arbuscular mycorrhizal fungi (AMF), the symbiotic mutualisms between the fungi and roots of higher plants [46], has been reported in the rhizosphere of many types of natural vegetation or plants in West Timor [6,47–49]. As the soils of West Timor are mostly calcareous and less fertile, the abundant indigenous mycorrhizal could become a biological natural supporting mechanism that could enhance plant growth in the region by improving plant nutrient absorption; most notably, phosphorus (P), which is highly adsorbed in calcareous soil. Moreover, mycorrhizal fungi are also known to improve plant resistance to drought [46]. This can help in the survival of plants under water stress commonly occurring on dryland in West Timor. Indigenous arbuscular mycorrhizal fungi (AMF) were found to be more abundant in the traditional farming system, where farmers perform minimum tillage and avoid the use of inorganic fertilizer, than the modern agriculture system, which involves the use of inorganic fertilizer [6]. This finding suggests that the traditional farming system in West Timor favors the health of soil biology. The potential of indigenous AMF to improve the soil phosphorus (P) availability and growth of maize with a lower amount of inorganic P in the calcareous soil of West Timor has also been recently reported [26].

The potential of indigenous plant-growth-promoting rhizobacteria (PGPR) has also been reported as a cost-effective and safe alternative approach to improve plant growth and developments, as they are able to produce plant-growth regulators, as well as enhance the soil condition and protection from soil pests and diseases [50]. In [51], it was found that indigenous *Bacillus* sp. and *Pseudomonas* sp. were capable of reducing brown rot gummosis disease in Soe Mandarin. *Bacillus* sp. has also been reported to produce indol acetic acid (IAA) and increase phosphorus solubility [52]. In [53], it was also found that local *Bacillus* spp. had the potential to control rice leaf spot disease caused by *Drecschlera oryzae*. Furthermore, this study also indicated that *Bacillus* spp. can also stimulate plant growth. The occurrence of other types of PGPR in West Timor included *Gluconacetobacter* sp. [54] and *Acinetobacter* sp., *Enterobacter* sp., *Klebsiella* sp., and *Pantoea* sp. [45].

Two other potential soil microorganisms that are also important in supporting plant growth in the semi-arid land of West Timor are *Trichoderma* and *Streptomyces*. *Trichoderma* was found to be ubiquitous in different plants in West Timor, such as mandarin, rice, maize, tomato, and chili. *Trichoderma* has been found to play a number of roles in agriculture, including promoting plant growth [55,56] and inducing resistance to plant pathogens [57–59]. Within the context of the West Timor dryland agro-ecosystem, indigenous *Trichoderma* was found to be able to restrict the growth of root and basal stem rot disease of soe mandarin caused by *Phytophthora palmivora* [60], and also to reduce the growth of *Diplodia* sp. in vitro [61]. In addition, indigenous *Trichoderma* species from West Timor were also found to suppress brown spot disease and increase the yield and yield-contributing characteristics of upland rice [62]. In a pot experiment conducted in [62], the local *Trichoderma* species propagated in corn rice bran were applied to the planting media before planting, while the fungicide was applied through spraying. The study results are presented in Table 2.

**Table 2.** The effect of local *Trichoderma* species and from West Timor on brown spot disease (*D. oryzae*) and the yield of upland rice cv. Inpago 7.

| Disease Control Treatment | AUDPC (%/Day) * | Efficacy (%) | Grain Yield per Plant (g) | A Grain Yield Increase (%) ** |
|---|---|---|---|---|
| *T. viride* | 913.69 [a] | 31.51 [a] | 30.7 [ab] | 23.5 |
| *T. harzianum* | 905.49 [a] | 32.12 [a] | 28.8 [a] | 16.1 |
| *T. hamatum* | 907.84 [a] | 31.94 [a] | 34.3 [c] | 38.3 |
| Fungicide Trivia 73 WP | 766.81 [a] | 42.52 [b] | 26.5 [a] | 6.70 |
| Control (without disease control) | 1333.97 [b] | | 24.8 [a] | |

Note: AUDPC: area under the disease progress curve. Lowercase denotes a comparison within the same column, and uppercase indicates a comparison within the same row. Values within the same column/row with the identical lowercase or uppercase are not significantly different based on the LSD (0.05) post hoc test. * Determined as the percentage of decrease in AUDPC from that of the control treatment. ** Determined as the percentage of increase in grain yield per plant from that of the control treatment. Source [62].

Table 2 shows that three local *Trichoderma* species from West Timor were able to reduce the brown spot disease severity by up to 31–32% as compared to the control, but the efficacy of these *Trichoderma* species was still below that of the fungicide Trivia 73 WP, which reached 42.5%. Interestingly, the grain yield increase caused by the three *Trichoderma* species treatments was much higher than that of the fungicide treatment. This finding suggests that *Trichoderma* applications not only suppressed the disease but also increased the grain yield of upland rice, presumably through the mechanism of plant-growth stimulation, as reported by previous workers [55,56].

*Trichoderma* used as fertilizer (mixed with compost/trichocompost) also reduced the fusarium wilt of tomatoes and increased tomato growth [63], as well as increasing the growth of edamame [64]. The above study results highlight the significant role of local *Trichoderma* isolates that can be optimally used in maintaining and increasing the soil biological health of West Timor, thus contributing to supporting the agriculture production in the region.

Besides arbuscular mycorrhizal fungi (AMF) and *Trichoderma*, both *Streptomyces pseudogriseolus* and *S. cellulosae* bacteria were found in maize; each has potential as a biological control for *Escherichia coli* and *Staphylococcus aureus*, and *Fusarium oxysporum*, respectively; both were found in maize rhizosphere [65].

In addition to these beneficial soil microorganisms, Taopan et al. [66] found Methanotrophic bacteria in West Timor (Kupang), namely, *Methylocystis rosea*, *Methylobacter* sp., *Methylocystis parvus*, and *Methylococcus capsulatus*, in paddy rhizosphere soil, which could increase rice production and decrease methane emission.

*2.3. Farming and Soil Fertilization*

Although there is various local wisdom practiced by farmers in West Timor, there is a counter-productive action, "the slash-and-burn system", which is currently still carried out by local farmers before planting. Since no study has been undertaken on the impact of slash-and-burn on the soil microbiota in West Timor, this review was based on similar studies conducted elsewhere. The effect of fire on soil microbes has been reported with various results. For instance, in [67], it was found that in a long unburnt site (45 years) of the *Eucalyptus marginata* forest, *ectomycorrhiza* levels were higher than those of the site that had remained unburnt for 6 years and 1 month. Accordingly, fire could not only eliminate the substrate of certain *ectomycorrhizas*, but could also have a sterilizing effect that could reduce the inoculum potential of the fungal symbionts [67]. In contrast, other authors found that the exclusion of fire from the Eucalyptus forest resulted in an adverse impact on *mycorrhizal* association due to soil ecological changes [68,69]. The impact of burning on soil microorganisms may be related to the time of burning [70]. The study found that the abundance and diversity soil microorganisms decreased at 2 and 4 weeks after burning, respectively, but increased 6 weeks after burning. This suggested that if the soil is left undisturbed for a long time after burning, the soil, including its microorganisms, may

recover [70]. However, the capability of soil microbiota to recover after burning might be different depending on the resistance of the soil microbes to high temperatures [70,71].

Slash-and-burn resulted in changes in soil chemical properties, including reducing soil acidity and increasing some base cations [72]. Following changes in the soil chemical properties due to slash-and-burn, there were also changes in some bacteria communities' taxa and some functional bacteria, and these changes could be considered a buffer for drastic changes in soil fertility after slash-and-burn [73]. Changes in soil chemical properties, such as a decrease soil organic C and N, as well as changes in soil biology, such as soil respiration, microbial biomass C and enzyme activities due to prescribed burning, have also been reported [18]. Despite the improvement of some soil chemical factors, such base cations, pH, and CEC due to slash-and-burn, it is a fact that soil microbes are the components of soil biology that are most affected. Thus, in the future, minimizing the frequency and intensity of the burns may need to be considered for a more sustainable farming system in West Timor dryland.

In addition to the potential of indigenous beneficial soil microorganisms found in the soil of West Timor, there are also some traditional agricultural cultivating systems practiced by local farmers that can maintain the biological health of soil, including minimum or zero tillage, as well as the use of green manure, cow manure, and a mixed-cropping planting system of leguminous plants, such as peanut, mungbean, or pigeon pea, together with non-leguminous plants, such as maize, pumpkin, and/or Siamese pumpkin. Minimum tillage or zero tillage is a conservative way to maintain land sustainability, including soil biological health [74,75]. This practice has been undertaken by farmers for a long time as a local technique to maintain soil fertility [38,76,77].

The use of manure as an organic fertilizer has also been practiced by local farmers. This approach can not only improve the biology properties of the soil but also its physical and chemical properties. The soil's physical properties, particularly soil structure, are very important in relation to the capability of the soil to hold more water and the resistance of soil to erosion. This becomes more crucial regarding the short rainy season in West Timor and the slope topography. The application of biochar and cattle manure in the semi-arid dryland farming system of West Timor has also been found to be beneficial to soil characteristics and to increase mungbean yield [78]. Another local wisdom of West Timorese farmers is mixed planting between leguminous and non-leguminous plants as an economic strategy to meet farmers' needs for food crops, as well as to avoid failed harvest. This local wisdom is not only beneficial from the economic perspective of the farmers, but is also ecologically beneficial for improving soil health through the use of leguminous plants. This practice has been undertaken by farmers for a long time as local wisdom to maintain soil fertility. The use of forage legumes in a mixed-cropping system with maize increased the soil nitrogen level and increased the maize yield substantially in West Timor soil conditions [79].

### 2.4. Crop Diversity in Upland Farming

Crop diversity including food crops, vegetables, fruits, and estate crops in West Timor is high among crop types and varieties. Crops are cultivated by farmers in West Timor based on altitude and water availability, as stated in [80,81]. Additionally, the water availability determines crop types and rotation patterns in a given region [82,83]. Crops that are cultivated at high altitude (>700 m asl) are different from those grown at low to middle altitudes (0–<700 m asl). Similarly, crops cultivated on land that has permanent water sources are different from those on dryland without water sources.

Crops cultivated at high altitudes in West Timor, such as in North Molo and Fatumnasi Subdistricts in South Central Timor District, and also in West Miomafo of the North Central Timor District, include maize (*Zea mays* L.), sweet potato (*Ipomoea batatas* L.), potato (*Solanum tuberosum* L.), carrot (*Daucus carota* L.), kidney-bean (*Phaseolus vulgaris* L.), prey onions (*Allium porrum* Leek), coriander (*Koriandrum sativum* L.), snaps (*Phaseolus vulgaris* L.), garlic (*Allium sativum* L.), mandarin (*Citrus reticulata* L.), apple (*Malus domestica* L.), and

coffee (*Coffea arabica* L.) [84]. Horticultural crops dominate in the highlands, and they are planted more by farmers to earn an income [85].

Crops cultivated on irrigated land at low to middle altitudes in West Timor include irrigated rice, maize, onion (*Allium cepa* L.), brassica (*Brassica chinensis* L.), chayote (*Sechium edule* (Jacq.) Sw.), long beans (*Vigna cylindrical* L.), bitter gourd (*Momordica charantia* L.), chili (*Capsicum annum* L.), tomato (*Solanum lycopersicum* L.), melon (*Cucumis melon* L.), water melon (*Citrullus lanatus* L.), grape (*Vitis vinifera* L.), and mungbean (*Phaseolus radiatus* L.) [84]. Most of the vegetable crops are farmed after the main crop rice is harvested or during the dry season.

However, crops cultivated on dryland at low to middle altitudes include upland rice (*Oryza sativa* L.), maize (*Zea mays* L.), pumpkin (*Cucurbita moschata* Durch), rice bean (*Vigna umbelata* Thunb), pigeon pea (*Cajanus cajan* L.), ground nut (*Arachis hypogaea* L.), mung bean (*Vigna radiata*), barley (*Setaria italica* L.), job's tear (*Coix lacrima-jobi* L.), sorgum (*Sorghum bicolor* L.), banana (*Musa* sp.), coconut (*Cocos nucifera* L.), mango (*Mangifera indica* L.), betel nut (*Areca catechu* L.), betel (*Piper betle* L.), salaka (*Salacca zalacca* L.) cassava (*Manihot utilissima* Pohl), and cashew nut (*Anacardium occidentale* L.). The crop diversity determines the soil fertility and farm sustainability in a given region. Cultivating various crops in a parcel of land, on the one hand, requires different nutrients from the soil, while on the other hand, provides a low crop failure risk.

Polyculture farming is considered local Timorese knowledge, and it involves farming or planting crops that have economic, social, and ecological benefits. Farmers in some villages of the Mutis highland deal with cultivated horticultural crops, such as onion (*Allium cepa*), garlic (*Allium sativum*), potato (*Solanum tuberosum*), chili (*Capsicum annuum*), maize (*Zea mays*), groundnut (*Arachis hypogaea*), cassava (*Manihot esculenta*), and sweet potato (*Ipomoea batatas*). Some farmers' groups cultivate some herbs and medicinal plants, such as ginger (*Zingiber officinale*), galangal (*Alpinia galanga*), aromatic ginger (*Kaempferia galanga*), turmeric (*Curcuma longa*), and Curcuma (*Curcuma zanthorrhiza*) [5,9].

Farmers simply let plants or vegetation grow naturally in the gardens of their homes and on upland farms, such as nutmeg (*Myristica* sp.); the ficus tree (*Ficus* sp.); the small cotton tree (*Bombax malabarica*); white-barked Acacia (*Acacia leucophloea*); the lac tree (*Scheilechera oleosa*); the betel nut palm (*Areca catechu*); albizia (*Albizia chinensis*); the helicopter tree (*Gyrocarpus americanus)*; as well as trees for bees: *Wenlandia buberkilli* var. Timorensis, *Todalia asiabeca,* and *Albizzia saponaria* [9]. They also plant or sell wooden vegetations for building materials, such as mahogany trees (*Swietenia machrophylla* King, *Swietenia mahagony* L. Jacg.), white teak (*Gmelina arborea* (Burm F.) Merr), teak (*Tectona grandis* L.), suren toon/iron redwood (*Toona sureni* (Blume) Merr), Timoo wood (*Timonius sericeus* (Desf) K. Schum), the bastard poon tree—*kepuh* (*Sterculia foetida* L), blackboard trees—*pulai* (*Alstonia scholaris* (L.) R.Br, *Alstonia spectabilis* R.Br), jackfruit trees (*Artocarpus heterophyllus* Lamk and *Artocarpus integra* Merr), and white-barked Acacia (*Acacia leucophloea),* which are found in the environments surrounding their settlements. Farmers also plant horticultural crops, such as pineapple (*Ananas comosus* Merr), soursop (*Anona muricata),* jackfruit trees *(Artocarpus communis* Forst, *Artocarpus heterophyllus* Lamk, *Artocarpus integra* Merr), achira (*Canna edulis* Ker), papaya (*Carica papaya* L.), pummelo (*Citrus maxima* (Burm) Merr), Taro—*ubikeladi* (*Colocasia esculenta* Schott), coconut (*Cocos nucifera),* asiatic yam (*Dioscorea aculcata* Linn and *Dioscorea alata* Linn), sweet potato (*Ipomoea batatas* Poir), mango (*Mangifera indica),* cassava (*Manihot utilissima* Pohl), banana (*Musa parasidiaca* Linn), avocado (*Persea gratissima* Gaertn), and turkey berry—*terung pipit* (*Solanum torvum* Swartz). Several NTFPs, such as betel nut palm—*pinang* (*Areca cathecu),* tamarind—*asam* (*Tamarindus indica),* and candle nut—*kemiri* (*Aleurites moluccana)* were developed around settlements, including plants that are useful for conservation, such as weeping fig—*beringin* (*Ficus benyamina)* and the blackboard tree (*Alstonia scholaris),* to increase the biodiversity, land conservation, and reforestation of areas around springs [86].

Based on the listed crops found in various places in Timor, most food crops were categorized as indigenous food crops cultivated for household consumption and income.

The crops were planted simply following seasonality, and government intervention was very limited. Seed or planting materials were prepared by farmers themselves from the previous harvest season [87,88]. Most grains and seeds were stored in a traditional house called a *Lopo*, which is normally used for cooking. The smoke from firewood in the *Lopo* house protects the quality of seed materials, at least up to the next planting season [88–90].

## 2.5. Plant biodiversity in Local Agroforestry of Mamar

Upland farming in Timor started generally from land clearing and planting food crops for subsistence. The intensification of farming led to an increasing number of crop species, including some natural vegetations, which later developed into the so-called local agroforestry or secondary forest of *Mamar*. There were 112 crops/vegetation species in *Mamar* identified from 16 different samples, which emphasizes the important roles of *Mamar* [91].

These crops have a multi-strata characteristic from seedlings, saplings, poles, and trees, which indicates an important value index (IVI). At the seedling level, 85 species plants were identified, 10 of which were categorized as dominant species, namely, *Pteris vitata*, *Piper betle*, *Chromobium* sp., *Chromolaena odorata*, *Corypha gebanga*, *Euphatorium odorata*, *Paspalum conyugatum*, and *Salanua verox*. Seventy-five species had an IVI of approximately 1.13 (*Albizia lebbeck*) up to 9.66 (*Musa paradisiaca*). Based on the crop density, 10 species were classified as dense, namely, *Piper betle*, *Pteris vitata*, *Leucaena leucocephala*, *Chromolaena odorata*, *Corypha gebanga*, *Euphatorium odorata*, *Paspalum conyugatum*, *Areca cathecu*, *Imperata cylindrica*, and *Cassia siamea*.

Ninety species were found at the *sapling* level, five of which were classified as dominant species, such as *Leucaena leucocephala* (19.63), *Litsea umbellifera* (13.36), *Arechacathecu* (13.19), *Chromolaena odorata* (11.42), and *Cananga odorata* (10.10). There were 85 species that had an IVI of approximately 1.21 (*Acacia auriculiformis*) to 8.94 (*Swietenia machrophylla*). Based on the density, 10 species were considered dominant, namely, *Leucaena leucocephala*, *Areca cathecu*, *Chromolaena odorata*, *Cananga odorata*, *Corypha gebanga*, *Macaranga tanarius*, *Aphanamixis polysticia*, *Alstonia scholaris*, *Sesbania grandiflora*, and *Jatropha curcas* [30].

At the pole level, 77 crop species were found, 12 of which were found to be dominant, namely, *Areca cathecu* (88.81), *Swietenia machrophylla* (20.58), *Calliandra calothyrsus* (18.56), *Cananga odorata* (13.88), *Artocarpus heterophyllus* (12.70), *Jatropha curcas* (12.61), *Antocephalus cadamba* (12.07), *Carica papaya* (11.69), *Coffea* sp. (10.71), *Leucaena leucocephala* (10.69), *Ficus albiphila* (10.40), and *Cocos nucifera* (10.04). Sixty-five species had an IVI of approximately 2.80 (*Acacia auriculiformis*) up to 9,67 (*Aleurites moluccana*). Further, based on the density, there were 10 species that were found to be dominant, namely, *Areca cathecu*, *Swietenia machrophylla*, *Cananga odorata*, *Artocarpus heterophyllus*, *Aleurites moluccana*, *Alstonia scholaris*, *Persea americana*, *Mangifera indica*, *Pterocarpus indicus*, and *Bambusa blumuena* [30].

Sixty-nine species were found at the tree level, among which 10 species were found to be dominant, namely, *Cocos nucifera* (84.75), *Alstonia scholaris* (19.96), *Aleurites moluccana* (19.35), *Ficus septica* (17.32), *Ficus benyamina* (15.38), *Musa paradisiaca* (15.24), *Mangifera indica* (14.74), *Toona sureni* (11.62), *Swietenia machrophylla* (11.30), and *Acacia leucophloea* (10.35). Fifty-nine species had an IVI of approximately 2.09 (*Bauhinia malabarica*) up to 9.91 (*Cananga odorata*). Based on the density, there were 10 species that were found to be dominant, namely, *Cocos nucifera*, *Aleurites moluccana*, *Mangifera indica*, *Acacia leucophloea*, *Swietenia machrophylla*, *Ficus benyamina*, *Artocarpus heterophyllus*, *Sterculia foetida*, *Tectona grandis*, and *Persea Americana* [30].

*Cananga odorata* is widely dominant in seedlings, saplings, poles and trees. *Areca cathecu*, *Leucaena leucocephala*, and *Swietenia machrophylls* species were dominant in saplings and poles, while some species had absolute dominance at certain levels. The vegetation dynamics of *Mamar* vary; dominancy of the specific crop at a certain level is not absolute. Dominance in poles and trees is not parallel to in seedlings and saplings; it is rather determined by the ability of each species to adapt and compete in using the resources needed for growth. *Cocos nucifera* has high dominancy at the tree level, with an IVI of 84.75,

and conversely, this is very low at the seedling level, with an IVI of 3.02 and a sapling level IVI of 8.56. A similar condition occurred for *Areca cathecu*, which is dominant at the pole level, with an IVI of 88.81, but low at the seedling level, with an INP of 8.05, and at the sapling level, with an IVI of 13.09.

Based on the Braun–Blanquet category [92], most of the vegetation in the *Mamar* community was found to be *occasional* and *frequent*; some were found to be in the *abundant* and very *abundant* categories, particularly for lower vegetation. For poles and trees, species abundance was included in the *rare* and *occasional* categories, while *Areca cathecu* and *Cocos nucifera* were included in the *abundant* and even highly abundant categories in some samples.

Seedlings and saplings have significant value in determining structure and composition, and they have a role as *trees of the future* to replace the *trees of the present*, and also as an indicator for a sustainable ecosystem. In contrast, the reality for the *Mamar* community is that species dominancy in the seedling and sapling groups is not parallel with dominancy in the pole and tree groups, including dominant species at the seedling and sapling levels, which is not important economically. Most of the dominant species in the seedling and sapling groups are superior species in competition with one another that adapt to the environment. These include *Salanua verox*, *Euphatorium odorata*, *Colocasia esculenta*, *Imperata cylindrica*, *Pteris vitata*, *Chromolaena odorata*, and *Piper betle* [30].

Local wisdom in managing crop diversity in the local agroforestry of *Mamar* is pivotal for *Mamar* sustainability. Farmers' intervention to diversify species determines the complexity of crop diversity in the *Mamar* ecosystem. Dominant species at the pole and tree levels are *Areca cathecu* and *Cocos nucifera*, which have high economic value; however, they are not dominant at the seedling and sapling levels due to frequent harvesting of these two species, which disturbs their regeneration process. On the other hand, maintaining species with low economic value in the seedling and sapling groups is a pivotal strategy to maintain sustainability and increase the productivity of the crops that have high economic value [30].

Information regarding the structure and composition of vegetation showed a high diversity of crops/vegetation in the local agroforestry of *Mamar*. The structure and composition of multi-strata vegetation from seedlings to trees indicated a high level of stability of *Mamar* as a living space and a regeneration process of vegetation in the semi-arid ecosystem. This stable structure and composition benefitted the improvement of environmental services. A challenge for long-term management is that maximal production function is difficult to reach because the *Mamar* community is not purely driven by economic goals, but also by social and ecological aspects, the conservation of cultural key species, and protecting water resources/springs. Therefore, ecological function as a supporting system for the productivity function, which has implications for economic function, can be applied to various vegetation to produce biomass at the surface layer [93].

- *Similarity Index on Agroforest Mamar*

Similarity analysis based on the important value index, species density, and environmental factor in every site showed variation at every growth level. The observed environment factors are pH, kalium (K), phosphorus (P), calcium (Ca), magnesium (Mg), natrium (Na), nitrogen (N), cation exchange capacity (CEC), saturation base, slopes, altitude, and rainfall. Overall, the similarity index tended to be strong when a number of environment variables were included in the analysis (Table 3).

**Table 3.** Dynamic changes in similarity index based on the important value index (IVI) and density of plants and environment factors (EFs) in the *Mamar* ecosystem.

| No. | Groups | Range of Index Similarity | | | |
|---|---|---|---|---|---|
| | | IVI (Plants) | IVI + EF | Density (Plants) | Density + EF |
| 1 | Seedlings | 8.5–59.44 | 37.80–71.55 | 9.10–57.87 | 39.54–72.02 |
| 2 | Saplings | 15.34–69.87 | 41.00–75.86 | 12.07–62.04 | 41.24–72.37 |
| 3 | Poles | 19.42–66.93 | 44.25–72.30 | 10.52–60.14 | 44.28–73.26 |
| 4 | Trees | 25.05–68.59 | 49.04–79.82 | 23.97–69.57 | 47.05–79.78 |

Source: Processed from a plant biodiversity survey on the *Mamar* ecosystem [30].

The similarity index (SI) showed variation in each sample at the seedling, sapling, pole, and tree levels based on the important value of the index parameter (IP) and vegetation density. Similarity tended to increase when a number of environmental factors were involved at every growth stage. The higher SI of each sample indicated the lower diversity management. This affects the output of ecological services in terms of soil fertility, habitat function, biomass, and carbon [94,95]. The semi-arid region with a marginal environment needs farmers' intervention by increasing crop species combinations to support a stable ecosystem [96,97].

*2.6. Natural Vegetation Biodiversity in Relation to Livestock Farming*

The diversity of vegetation not only in the farmland in *Mamar*—as previously mentioned, but also in the forest and grassland—supports integrated livestock farming in the region. Although livestock may consist of cattle, buffalos, horses, goats, sheep, pigs, and chicken, the most important practice in West Timorese is raising cattle. The general vegetation biodiversity—as well as crop and forest vegetation biodiversity—that supports the integration of livestock farming in West Timor includes the natural grassland ecosystem (consisting of native grasses and herbaceous legumes), native trees (consisting of legume and non-leguminous trees) in the savannah, *Ladang* (upland), or in the form of forests in *Mamar*. Native grass species in the native grasslands in this case include *Heteropogon contortus*, *Ischaemum timorense*, *Sorghum timorense*, *Sorghum nitidum*, *Cenchrus polistachyon* (syn.: *Pennisetum polystachion*), *Rottboellia exaltata*, and *Bothriochloa pertusa*. *Cenchrus polystachion* [98] and *Rottboellia exaltata*, especially, are annuals, which are usually cut and carried during the early wet season before they start to flower and are fed to cattle in the pen, playing an important role in fodder composition in West Timor for fattening cattle, though in many places these two annuals are considered weeds [99–103]. Other grasses are usually free grazed or fed to the animals by tethering them in communal native pastures. Several herbaceous legumes identified as important fodder components in native pastures included *Aeschynomene americana*, *Alysicarpus vaginalis*, *Desmodium timorense* (a broad leaf legume), *Mucuna timorense* (similar to *M. pruriens*, but it is more hairy and itchy), and *Desmanthus virgatus*. There are some naturalized species which were introduced into Indonesia a long time ago as cover crops in the estate crop plantations, including *Macroptilium atropurpureum* (Siratro), *Centrosema molle* (Syn.: *Centrosema pubescens*), and *Calopogonium muconoides*, which are found to grow naturally in native pastures, at road sides, in *Ladangs*, and at the edge of forests and bushes. *Desmodium timorense* (the broad leaf *Desmodium* in Timor) can be found sporadically in spots of native grasslands and are grazed by free-grazing animals (both goat and cattle); however, in the North Central Timor district, farmers cut and carry this as feed to fatten cattle [104,105].

The native legume trees found in West Timor include *Acasia leucophloea* and *Acasia nilotica*, which may be grazed by free-grazing cattle and goats when the plant sizes are within the reach of the animals, while the tall-growing plants may be climbed and cut down by the farmers to feed to the animals. The leaves of *A. leucophloea* contain a moderate–high protein content of 15–17% [106]; they are also an important source of dry season fodder and pasture trees in Pakistan, with 25% crude protein content in seeds [107], and of plant nectar for honey bees in Timor [9]. Some naturalized species that have been present for quite

a long time include *Glirisidia sepium*, *Sesbania grandiflora*, and *Leucaena leucocephala* subsp. *leucacephala* (small common *Leucaena*) [98]. These small trees may be directly grazed by the animals or cut and carried as feed for animals in pens or at tethering places near to farmhouses. Both of the Acacias are commonly cut and carried during the long dry season on the island. The small type of Common Leucaena has been commonly used in the farming systems, especially in the Amarasi areas, in West Timor, as an effort to prevent *Lantana camara* infestation in the area, to improve the soil quality, and as a pioneer plant in the slash-and-burn method to shift cultivation. The plant is grown at a high density in a plot of land and will be cut down and burnt (slash-and-burn) in the land preparation season (before the wet season) before being planted with maize in the wet season. After this, the plant will be left to regrow and cover the land plot, and will be the same at the next land preparation stage, while during this time, the plant may be cut and carried as feed to fatten cattle near the farmhouses. Later, the introduction of the giant leucaena varieties (*Leucaena leucocephala* subsp. *glabrata*) in the 1970s, such as K8, K28, and K 500 (cv Cunningham), enriched the fodder sources of livestock farmers in West Timor, especially in the Amarasi area [98]. However, upon the attack of the psyllid insect (*Heteropsylla cubana*), a new variety (*Leucaena leucocephala* cv Tarramba) that was resistant to the insect was tested, promoted, and developed, especially for ruminant feeding [105,108].

Besides the tree legumes, there are some native trees of non-leguminous plant leaves that are important in providing highly nutritional fodder, especially during the mid to end of the dry season (August to November/December), for the livestock (cattle and goats), e.g., *Macaranga tanarius* (local name: *Busi*), *Schleichera oleosa* (local name: *Kesambi*), *Ceiba petandra* (lokal name: *Kapok*), and *Ficus* species (local name: *Beringin*). These trees are important during the dry season when grasses and herbaceous legumes are scarce. These non-leguminous trees, besides providing fodder during the dry season, and timber (or large to medium branches/trunks) obtained from logging the trees, as well as other trees such as Gum Trees (Eucalyptus species), would be important in building fences for food crop land plots during the planting season to prevent free-grazing animals from damaging the cultivated plants within the plot. Besides the mentioned feed obtained from native grasslands, food crop waste, and trees from forests and *Ladangs*, especially as a source of protein and fiber for ruminants, the *Corypha gebanga* (local name: *Gewang*) pith has been quite an important feed as a readily available carbohydrate (RAC) or energy source included in the local wisdom of West Timorese farmers for livestock farming. The pith of the *C. gebanga* can provide feed for cattle, goats, pigs, and chickens [109,110].

### 2.7. Farmers and Farming Environment

Based on the adoption of agricultural trends, the indigenous farmers of Timor have been known to find it difficult to embrace new agricultural innovations. The level of agricultural adoption, in general, is very low, which is closely related to the intrinsic semi-arid farming environment. This is also closely related to household characteristics, in that most of the farmers do not finish primary school (almost 70%), and some do not finish junior and high school (25%) [111].

The household structure indicated that there were 243,635 households with an average number of household members, with 27% of them working as farmers (Table 4), which indicated that the number of household members working in the agriculture sector was very low, while other members had not yet entered the workforce. These data are crucial, since family size is related to the contribution to the workforce, income, and social and economic burden [112].



**Table 4.** Number of household farmers in West Timor, ENT.

| Regency/City | Number of HH Farmers | Household Member | | | | Number of Farmers | | | |
|---|---|---|---|---|---|---|---|---|---|
| | | Man | Women | Total | Average HH Member | Man | Women | Total | % |
| Kupang | 59,601 | 122,624 | 122,151 | 244,775 | 4.11 | 53,815 | 12,353 | 66,168 | 27.03 |
| TTS | 99,877 | 183,627 | 187,716 | 371,343 | 3.72 | 84,476 | 21,938 | 106,414 | 28.66 |
| TTU | 45,913 | 91,085 | 94,920 | 186,005 | 4.05 | 39,935 | 9739 | 49,674 | 26.71 |
| Belu | 25,677 | 57,975 | 59,600 | 117,575 | 4.58 | 21,903 | 7841 | 29,744 | 25.30 |
| KupangTown | 12,567 | 29,772 | 29,379 | 59,151 | 4.71 | 10,700 | 2433 | 13,133 | 22.20 |
| West Timor | 243,635 | 485,083 | 493,766 | 978,849 | 4.02 | 210,829 | 54,304 | 265,133 | 27.09 |

Note: HH: households. Source: Survey of Agriculture and Livestock 2017 [111].

Table 5 shows that 76% of households received income from the agricultural sector, which is probably due to the following reasons: (a) workforce working outside the agricultural sector; (b) disguised employment; (c) young family structure; (d) more hired labor for farming activities; and (e) it has a low impact on conservation efforts and maintaining soil quality.

**Table 5.** Income source of households in West Timor, ENT.

| District | Number of Household | Income | | | | Number of Household Own Land | Number of Household Own No Land |
|---|---|---|---|---|---|---|---|
| | | Farming | % | Non-Farming | % | | |
| Kupang | 59,601 | 45,182 | 75.81 | 14,419 | 24.19 | 59,072 | 30,755 |
| TTS | 99,877 | 85,642 | 85.75 | 14,235 | 14.25 | 99,874 | 54,447 |
| TTU | 45,913 | 32,800 | 71.44 | 13,113 | 28.56 | 45,901 | 16,621 |
| Belu | 25,677 | 18,009 | 70.14 | 7668 | 29.86 | 25,469 | 11,095 |
| Kupang town | 12,567 | 2827 | 22.50 | 9740 | 77.50 | 11,611 | 10,605 |
| West Timor | 243,635 | 184,460 | 75.71 | 59,175 | 24.29 | 241,927 | 112,918 |

Source: Agricultural Survey 2018 [111].

Almost all farmers (99.3%) in West Timor have their own farming land; however, close to half (47%) of them own a narrow piece of agricultural land and can be categorized as small farmers. Both less labor force working in farmland and small land size have negative implications for soil quality and farming environments. Small farmers with small land sizes tend to maximize crop production and are taken into consideration less in relation to environmentally sound aspects.

Most farmers in Timor practice mixed-cropping patterns as a strategy to minimize risk [113]. They plant any suitable crops on a parcel of land and, at the same time, raise livestock as an important part of the farming system [114]. Household farmers normally own food crop-farming land, horticulture, and estate-crop land. Food crops are normally cultivated in rice fields (lowland and/or rainfed) and small areas in the gardens of their homes. Horticulture crops, particularly vegetables, are cultivated in the upland and in the rice field after the rice is harvested and if there is enough water available. Estate crops are normally planted in the local agroforestry plot, so-called *Mamar.* Upland farmers in Timor have employed soil conservation techniques, such as terracing and planting conservation vegetations [113].

### 2.8. Local Wisdom on Soil Management

Prior to the arrival of the European ruler in Timor, the indigenous Timorese survived as hunters and gatherers on the less populated island and in the diverse environment [115,116]. Timorese people probably started farming in the 13th century [117], and incorporated some new food crops, which were later replaced by maize after contact with Indian and Chinese traders, and later with European traders [115,118,119]. In this section, a pearl of local wisdom refers to indigenous Timorese practices in upland farming that respect environmental and sustainability notions.

Most of the agricultural land on Timor Island is considered marginal or unfertile land/soil [118] compared to Indonesia's western part. Coupled with low and erratic rainfall, farming, particularly food crop agriculture, encounters high risk and low productivity. Within this environment, Timorese farmers have developed farming strategies to maintain a level of food crop production for subsistence. Maintaining land productivity by local farmers is closely related to farm management, crop diversity, and crop residue management [120], while soil fertility is closely related to the length of the fallow period and succession vegetation [121].

There are three upland farming types on Timor Island [85]: (1) swidden agriculture (*kebun/ladang*), (2) local agroforestry (*Mamar*), and (3) house garden/farmyard (*pekarangan/kintal*). The first includes permanent plot cultivation (*ladang* permanent) and swidden plots or fallowed *ladangs*. The second refers to the dense mix of perennial crops and any other compatible plants in a relatively small parcel of land. It is considered the most stable, economic, and ecologically sound system. The last (house garden/farmyard) refers to the farmed land around or near farmers' homes. These three types of farming reflect maximum use and compatible resources and minimize agriculture risk in marginal areas. Diversification is also a form of self-insurance [122–124], reducing vulnerability [125,126] and improving resilience [125–127].

There are five main indigenous ethnic groups settled in the western part of Timor Island, i.e., Meto, Tetun, Bunak, Kemak, and Marae [119]. The Meto ethnic group settled and dominated the western part of the island. Tetun mostly settled in the southern part of the Malaka District. Three other ethnicities settled in the Belu and Malaka districts. Although they share a common farming practice, these ethnic groups developed specific strategies to survive in their local environment.

Timorese farmers investigated suitable land for crops based on the vegetation and population density. They perceived that the denser the vegetation, the more fertile the soil. Upland farmers started land clearing for farming if they perceived that soil was productive enough to support agriculture for several years before it was fallowed to allow for natural revegetation. Besides vegetation, farmers also observed earthworm secretion as an indicator of the soil fertility.

An ancient food crop commodity in Timor was foxtail [117], followed by the introduction of a new food crop, Maize, which later became the main staple. The slash-and-burn system is a common practice in land preparation for a traditional system. There is almost no plowing—or, in other words, farmers practice minimum soil disturbance. Upland farming for food crops is conducted mainly on sloping land; therefore, minimum soil disturbance is considered to minimize soil erosion and promote the quick recovery of soil biology.

Traditional farming practices had little impact on the natural ecosystem of Timor Island, at least up to the beginning of the twentieth century. Farmers strictly controlled fire in swidden agriculture to avoid wildfire [5]. During land clearing, some vegetation remain uncut, primarily foraging three legumes. Leucaena (*Leucaena leucocephala*) trees are the most common and widely used for cattle feed and fertile soil recovery in swidden agriculture. This typical swidden cultivation is considered a productive system for at least forest–grassland succession and maintaining biodiversity [128].

To minimize the risk in food crop farming in Timor Island's marginal areas, farmers practice a mixed-cropping pattern. To maintain or reduce soil fertility depletion, farmers have to combine different food crops that support one another or minimize competition in nutrient intake. Three main widely planted food crops are maize (*Zea mays*), pumpkin (*Cucurbitaceae*), and pigeon pea (*Cajanus cajan*). The composition of the three food crops depends on farmers' preferences in considering land and climate prediction. These three food crops also reflect the main diet of the Timorese people.

## 3. Discussion

Timor Island is one of the semi-arid regions mentioned in [129] that needs to be carefully considered by agricultural policymakers when implementing changes. The

recognition of the marginality of Timor Island was realized through farming system research conducted in the mid-1980s [5,130,131]; however, this research was more focused on increasing crop productions, while few attempts were conducted to explore the nature of the semi-arid region and indigenous knowledge in the land management of Timorese farmers.

Local wisdom practiced in the farming activities in West Timor is a response to the marginality of the dominant environment in semi-arid areas. Actions taken, such as combining different plants or food crops in a parcel of land, have proven to be the best strategy to maintain a sufficient level of food production and, at the same time, maintain soil biology and fertility [132–134]. Different legume crops, both in the upland and local agroforestry of *Mamar*, as mentioned in [135–137], provide rich soil biology [138]. From this local wisdom perspective, the notions of national agricultural policy that are biased toward monoculture and chemical inputs for crop production need to be adjusted when applied in the marginal semi-arid area of Timor Island.

The diversity of vegetation in West Timor not only provides benefits for the farming systems that are suited to its semi-arid climate and relatively young soils, but the community also supports the sustainability and development of integrated livestock farming in the region. Raising different kinds of livestock that suit the agro-ecological zones is pivotal in managing synergism in the crop–livestock system.

A mixed-cropping pattern is practiced mostly to minimize risk, maintain production at least for the subsistence level, and maintain crop diversity. Farmers plant any suitable crops in a parcel of land and, at the same time, raise livestock as an important part of the farming system.

Plant production largely depends on the genotype, year, and their interaction. Profitable plant production, based on high yields, is impossible without the application of proper cultivation technology [139–141]. In low-fertility soils, the application of fertilizers is necessary, but sowing legumes is also important due to its role in improving nitrogen levels in soil [140,141].

## 4. Conclusions

Sustaining environmentally sound farming not only requires proper input, but also a deep understanding of the existing natural characteristics of the region and the indigenous knowledge practices. To overcome these unfavorable conditions, over hundreds of years, farmers have established local wisdom in managing specific sustainable farming systems. With a population increase on the island, the size and quality of land owned by farmers continues to decline, which also affects the common local wisdom practices, and further investigation is warranted regarding sustainability.

The presence of soil microorganisms in marginal semi-arid soils is considered important to maintain soil health and fertility, as well as to support plant growth. Indigenous farming practices of Timorese farmers, in general, are actually preserving the existence of microorganisms to some degree to produce a subsistence level of sustainable and environmentally sound crop production.

Local wisdom-related farming enables farmers in Timor to effectively respond to the marginal environment of Timor Island. They embrace environmentally sound farming practices in land preparation, diversify crops/vegetation via a mixed-cropping pattern, and raise livestock in a crop–livestock system. Although they are quite vigilant toward the introduced agricultural innovations, they are actually willing to embrace any agricultural innovations that suit their household circumstances and environments.

It is clear that the increasing human population requires more land to be cultivated since the traditional system cannot support high crop productivity. External agricultural interventions to increase crop productivity that undermine the stability of the farming systems and threaten the survival of local microorganisms and soil decomposers should be re-evaluated.

**Author Contributions:** Y.N. oversaw the review and prepared the early manuscript based on local wisdom. T.B. and A.S. drafted soil-related aspects. G.N.N., E.P. and E.Y.H. drafted the farming system and local agroforestry of Mamar. L.I., A.V.S., Y.S.M. and N.R.E.K. drafted soil biology. J.N. and D.K.H. drafted forage and livestock. B.d. and H.d. drafted socio-economic aspects of the farming system. All authors have read and agreed to the published version of the manuscript.

**Funding:** The publication of the manuscript was financed by the Governor of East Nusa Tenggara.

**Institutional Review Board Statement:** Not applicable.

**Informed Consent Statement:** Not applicable.

**Data Availability Statement:** Not applicable.

**Acknowledgments:** This paper was developed and inspired by all authors' involvement in agricultural development and land protection/conservation in marginal semi-arid areas of Timor Island. We would like to thank the East Nusa Tenggara Assessment Institute of Agriculture Technology chairman, Nusa Cendana University, and Environment and Forestry Research and Development Institute of Kupang for allowing the authors to use resources in the respective institutions in the preparation of this manuscript. We would also like to thank the Governor of East Nusa Tenggara for financial support in the publication of this manuscript. All remaining errors are the authors' responsibility.

**Conflicts of Interest:** We declare that there is no conflict of interest.

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
