# Peer review of "Local Wisdom of West Timorese Farmers in Land Management"

_sustainability, doi:10.3390/su14106023_

Round 1

Reviewer 1 Report

The general remarks in the paper are that the paper was much clearer:

1. Please delete text in line 21-24 and add concrete results.

2. Please in line 457 see table 3 and table 4 in line 551. Tables 3 and 4 moves to the second pages, please set it to be on one sheet. Table 3. Changes Dynamic of similarity index based on the Important Value Index (IVI), Density of 457 Plants and Environment Factors (EF) in Mamar Ecosystem.
Table 4. Number of household farmers in west timor, ENT. 551

3. Improve the conclusion. For example: Move the text from line 658-663 to the discussion. The conclusions do not put citations. Add your conclusions.
Delete Line 658-663. Timor Island is one of the semi-arid regions mentioned by [128] that needs to be 658 carefully considered by agricultural policymakers in bringing changes. Recognition of 659 marginality of Timor Island has just been realized through farming systems research 660 conducted in the mid-1980s [77,129,130]; however, those research have more focus on 661 increasing crops productions, while few attempts were conducted to explore the nature 662 of the semi-arid region and indigenous knowledge in land management of Timorese.

4. Add citations in reference numbers 131-133.

5. Eject old references, 41-Doran Parkin,1996. You have too many citations and references, in total 130, please reduce them. For example. In Line 43-44. The mosaic environment, including the diversity of vegetation/plants and local microorganisms, are the manifestation of local soil and climatic conditions [4–14]. Delete ref. 5-14.

6. You have 36 omissions in the cited references (numbers 1, 2, 4, 17, 19, 26, 41, 44, 45, 50, 51, 52, 58, 59, 60, 61, 71, 75, 77, 78, 79, 80, 83, 90, 96, 97, 102, 103, 104, 105, 106, 112, 119, 124, 126, 128 and 130). Please correct the mistakes. Somewhere the year is missing, somewhere it is not bolded, or the Volume and number have been added.

7. Add citate and next references:

Citates: “Plants production largely depends on the genotype, year and their interaction. Profitable plant production, based on high yields, is impossible without the application of proper cultivation technology [131-133]. On low-fertility soils, the application of fertilizers is necessary, but sowing legumes is also important because of its role in improving nitrogen
levels in the soil [132, 133].

References:

131. Kolarić, Lj., Popović, V., Živanović, Lj., Ljubičić, N., Stevanović, P., Šarčević Todosijević, Lj., Simić, D., Ikanović, J. . Buckwheat Yield Traits Response as Influenced by Row Spacing, Nitrogen, Phosphorus and Potassium Management. Agronomy, 2021, 11, 12, 2371.
132. Popović, V., Vučković, S., Jovović, Z., Ljubičić, N., Kostić, M., Rakaščan, N., Glamočlija- Mladenović, M., Ikanović, J. Genotype by year interaction effects on soybean morphoproductive traits and biogas production. Genetika, Belgrade, 2020, 52, 3: 1055-1073.
133. Jovanović-Todorović, D., Popović, V., Vučković, S., Janković, S., Mihailović, A., Ignjatov, M., Strugar, V., Lončarević, V. Impact of row spacing and seed rate on the production characteristics of the parennial ryegrass (Lolium parenne L.) and their valorization. Notulae Botanicae Horti Agrobotanici Cluj-Napoca. 2020, 48, 3: 1495-1503

8. Please correct the omissions in order to improve the Paper. Thanks.

Author Response

Thank you very much for your comments and suggestions.

We do improve the manuscript, and re-organized the logical flows of the Manuscript as follows:

  1. We rewrite the Abstract as suggested.
  2. We do reorganize all the Tables and figure to be on one or similar page.
  3. We do improve the Discussion and Conclusion.
  4. The References improved by correcting the mistakes, deleting old references, and adding some relevant references. Unfortunately, we are unable to reduce the number of references since it is related to the additional information/improvement of the manuscript. 
  5. The details improvements are shown in the Manuscript (colored and track changes).

We hope that the improvement of our manuscript would satisfy you and meet the journal’s requirements. 

Reviewer 2 Report

Comments and Suggestions for Authors

In the current manuscript, titled “Local Wisdom of West Timorese Farmers in Land Management The authors appear to have done some interesting work and put in a lot of effort. However, there are some concerns. So, I suggest that the current version of the submitted manuscript be carefully revised and upgraded.

I also have a few other concerns, which I've described below.

  1. The “Title” of the article is appropriate and there is no need for any change.
  2. All the selected relevant aspects have been well placed under the respective sub-title of the article.
  3. The abstract opening sentence needs strong justification.
  4. The abstract is not well written; it is only a mere conscript of the study. Better would be to give some introduction followed by the gap in knowledge. The abstract is the only part of the paper that the vast majority of readers see. Therefore, it is critically important for authors to ensure that their enthusiasm or bias does not mislead the reader.
  5. Introduction: The introduction is poor. It does not reflect the aim; relevant literature and correlation of this study.
  6. What’s the gap of knowledge? Which is the scope of the manuscript?  The introduction should be revised accordingly.
  7. In "Introduction" and "Discussion", the authors should cite recent references between 2018-2021 from JCR journals.

Author Response

Thank you very much for your comments and suggestions.

We do improve the manuscript, and re-organized the logical flows of the Manuscript as follows:

  1. We rewrite the Abstract as suggested.
  2. We do improve/revise the Introduction as suggested.    
  3. We do improve the Discussion and Conclusion.
  4. The References improved by correcting the mistakes, deleting old references, and adding some new relevant references.
  5. The details improvements are shown in the Manuscript (colored and track changes).

We hope that the improvement of our manuscript would satisfy you and meet the journal’s requirements.